# Flea-Borne Typhus Causing Hemophagocytic Lymphohistiocytosis: An Autopsy Case

**Divya Chandramohan** [1], **Moyosore Awobajo** [2], **Olivia Fisher** [2], **Christopher L. Dayton** [3] **and Gregory M. Anstead** [1,4,*]

1. Division of Infectious Diseases, Department of Medicine, University of Texas Health San Antonio, 7703 Floyd Curl Drive, San Antonio, TX 78229, USA
2. Department of Pathology, University of Texas Health San Antonio, 7703 Floyd Curl Drive, San Antonio, TX 78229, USA
3. Division of Critical Care, Department of Medicine, University of Texas Health San Antonio, 7703 Floyd Curl Drive, San Antonio, TX 78229, USA
4. Medical Service, South Texas Veterans Health Care System, 7400 Merton Minter Blvd, San Antonio, TX 78229, USA
* Correspondence: anstead@uthscsa.edu

**Abstract:** Infection with members of the order Rickettsiales (the genera *Rickettsia*, *Anaplasma*, *Orientia*, and *Ehrlichia*) is known to cause hemophagocytic lymphohistiocytosis (HLH). The literature is scant on flea-borne typhus (FBT) being implicated in this process. We present a case of autopsy-proven HLH caused by FBT in a 71-year-old diabetic female who was initially suspected of having diabetic ketoacidosis who rapidly suffered decompensated multi-organ failure. Although she was suspected of having FBT and HLH pre-mortem, due to her rapid progression to multi-organ failure, she was transitioned to comfort care by her family five days after admission. A literature search yielded five other cases of HLH secondary to FBT, which are analyzed in this review. The literature on HLH occurring with infection due to other members of the order Rickettsiales is also surveyed.

**Keywords:** flea-borne typhus; hemophagocytic lymphohistiocytosis; fatal; *Rickettsia typhi*; *Rickettsia felis*; autopsy





## 1. Introduction

Flea-borne typhus (FBT) is an infection caused by the bacteria *Rickettsia typhi* and *R. felis*. It is typically an acute undifferentiated febrile illness, but about one-quarter of patients suffer respiratory, neurologic, hematologic, renal, hepatic, cardiac, ocular, or other complications [1]. About one-third of adult patients stricken with FBT require intensive care [2]. The infection is transmitted to humans by a flea bite or by the inoculation of a bite site, a skin abrasion, or mucous membranes with feces from fleas infected with these rickettsiae [3,4].

FBT is the most prevalent and widely distributed rickettsial infection, and it occurs on every continent except Antarctica [3]. In the United States, FBT is now relatively uncommon, but foci still exist in Texas, Hawaii, and California [5]. In the last decade, the incidence of FBT has increased in both Texas and California [6,7]. Outside of the United States, FBT is re-emerging in multiple locations.

The aim of the current study is to present an autopsy proven case of HLH secondary to FBT, and to provide a literature review of HLH due to infection by members of the order Rickettsiales (the genera *Rickettsia*, *Orientia*, *Ehrlichia*, and *Anaplasma*). Our review found five other cases of HLH attributed to FBT; one of the five patients succumbed to the condition. Early recognition of both FBT and HLH and rapid administration of doxycycline and immunosuppressive medication is paramount for increasing the chances for survival.

## 2. Case Presentation

A 71-year-old female with diabetes mellitus type 2, hypertension, and dyslipidemia was brought by her daughter during spring of 2021 to her primary care clinic (Day (D) 1) for confusion, generalized weakness, nausea, and increased urinary frequency of half a day duration. Her diabetes was managed with insulin glargine, empagliflozin, linagliptin, metformin, and liraglutide. A home blood glucose check in the afternoon was reported as 314 mg/dL. She was transported to the hospital and was lethargic on exam. Her physical exam was otherwise unremarkable. Additional history revealed that the patient resided in urban San Antonio, (TX, USA), and her housing was in poor condition with potential exposure to rats, opossums, and raccoons. She had no pets of her own and had no outdoor activities or known hobbies. Her triage vitals were significant for a respiratory rate of 26 breaths per minute and a blood pressure of 163/63 mm Hg. Initial laboratory evaluation revealed a white blood cell (WBC) count of 9.8 K/µL (reference range (RR) 3.4–10.4 K/µL), with a neutrophil count 8.8 K/µL (RR 1.5–6.6 K/µL), hemoglobin 14.2 g/dL (RR 11.5–14.9 g/dL), platelets 125 K/µL (RR 140–370 K/µL), bicarbonate 13 mmol/L (RR 20–29 mmol/L), anion gap 22 mmol/L (RR 8–12 mmol/L), glucose 148 mg/dL (RR 60–100 mg/dL), aspartate aminotransferase (AST) 153 U/L (RR 13–39 U/L), alanine aminotransferase (ALT) 55 U/L (upper limit of normal (ULN) 36 U/L), alkaline phosphatase 48 U/L (RR 45–117 U/L), total bilirubin 0.8 mg/dL (RR 0.2–1.2 mg/dL), albumin 3.0 g/dL (RR 3.2–5.0 g/dL). She was diagnosed with euglycemic diabetic ketoacidosis and was started on ceftriaxone, intravenous fluids, and an insulin drip. A right upper quadrant sonogram showed an echogenic liver consistent with steatosis/fibrosis, but no focal hepatic lesions were identified. A viral hepatitis serologic screen for hepatitis A, B, and C was negative. Her hemoglobin A1c was 7.1% (RR 4.0–6.4%). On D3, her anion gap normalized. She was febrile to 38.8 °C; two sets of blood cultures and a urine culture were negative. On D4, her fever persisted, and a computed tomography (CT) scan of the chest, abdomen and pelvis was performed, which was only significant for a lingular consolidation. A respiratory viral panel (nasopharyngeal swab; Biofire RP2.1 Assay®) and a *Legionella* urine antigen were negative. The C-reactive protein was elevated at 247 mg/L (ULN 10 mg/L). On D5, she had increasing levels of liver enzymes (AST 371 U/L, ALT 92 U/L, and alkaline phosphatase 111 U/L). Total bilirubin was normal (0.9 mg/dL). Fibrinogen was mildly decreased at 144 mg/dL (RR 152–445 mg/dL). On D6, she developed worsening mental status, increasing lactic acid to 5.3 mmol/L (RR 0.5–2.0 mmol/L), and hypotension unresponsive to intravenous fluids; she was transferred to the intensive care unit. She developed acute hypoxemic respiratory failure and was intubated. Norepinephrine and broad-spectrum antibiotics cefepime, metronidazole, vancomycin) were initiated. Results of laboratory testing on D6 showed WBC 23.2 K/µL, hemoglobin 11.3 g/dL, platelets 74 K/µL, and procalcitonin increasing from a D4 value of 2.74 ng/mL (RR ≤ 1 ng/mL) to >200 ng/mL. On D6, the lipase level was elevated at 825 U/L (RR 73–350 U/L). On D7, AST was 3361 U/L, ALT 392 U/L, alkaline phosphatase 403 U/L, and total bilirubin 2.4 mg/dL. Further evaluation included normal thyroid function tests and troponin-I level and a negative human immunodeficiency virus serologic test. A bronchoalveolar lavage was performed and submitted for bacterial, mycobacterial, and fungal cultures, all of which were negative. She had worsening lactic acidosis to 20.3 mmol/L and increasing pressor requirements and thus antibiotics were broadened to include doxycycline. Soon thereafter, the patient's family elected to transition her to comfort care. Prior to terminal extubation, laboratory findings were a hemoglobin of 8.1 mg/dL, platelet count of 64 K/µL, and elevated levels of lactate dehydrogenase at 3374 U/L (RR 92–240 U/L), d-dimer (58,663 ng/mL; RR < 500 ng/mL), and triglycerides (282 mg/dL; ULN 150 mg/dL). A ferritin level was not obtained. Her laboratory findings over the course of her illness are summarized in Table 1.

**Table 1.** Laboratory values during hospital course of the case patient.

| Test | D1 | D2 | D3 | D4 | D5 | D6 | D7 | D7 |
|---|---|---|---|---|---|---|---|---|
| WBC (K/μL) | 9.84 | - | 6.08 | 5.43 | 5.2 | 8.14 | 23.23 | 15.9 |
| Neutrophils (K/μL) | 8.75 | - | 5.57 | 4.69 | - | 6.76 | 20.89 | - |
| Hemoglobin (g/dL) | 14.2 | - | 12.8 | 13.9 | 12.9 | 12.1 | 11.3 | 8.1 |
| Platelets (K/μL) | 125 | - | 77 | 55 | 22 | 32 | 74 | 64 |
| Creatinine (mg/dL) | 0.77 | 0.72 | 0.35 | 0.42 | 1.18 | 1.65 | 2.59 | 2.90 |
| AST (U/L) | 153 | - | - | 310 | 371 | 1531 | 3361 | 2506 |
| ALT (U/L) | 55 | - | - | 87 | 306 | 350 | 392 | 310 |
| Alkaline phosphatase (U/L) | 48 | - | - | 89 | 111 | 317 | 403 | 311 |
| Total bilirubin (mg/dL) | 0.8 | - | - | 0.9 | 0.9 | 1.1 | 2.4 | 1.9 |
| Albumin (g/dL) | 3 | 2.3 | 2.4 | 2.3 | 2.2 | 1.7 | 1.5 | 1.1 |
| Procalcitonin (ng/mL) | - | 2.3 | - | 2.74 | - | >200 | - | - |
| Lactic acid (mmol/L) | 1.1 | 1.2 | - | 1.0 | - | 5.3 | 12.6 | 20.3 |
| LDH (U/L) | - | - | - | - | - | - | 1182 | 3374 |
| CRP (mg/L) | - | - | - | 261 | 247 | - | - | - |
| D-dimer (ng/mL) | - | - | - | - | - | 78,468 | 58,663 | - |
| Triglyceride (mg/dL) | - | - | - | - | - | - | 282 | - |
| Fibrinogen (mg/dL) | - | - | - | 61 | 144 | - | - | - |
| INR | - | 1.1 | - | - | - | 1.8 | 1.4 | - |
| Troponin-I (ng/mL) | <0.015 | - | - | - | - | 0.597 | 2.77 | 2.82 |

Abbreviations: Day 1, D1; White blood cells, WBC; Aspartate aminotransferase, AST; Alanine aminotransferase, ALT; Lactate dehydrogenase, LDH; C-reactive protein, CRP; International normalized ratio, INR.

From Table 1, an abrupt decrease in hemoglobin, platelet count, and albumin level is apparent by days 3–5. Furthermore, there was rapid increase in transaminase, creatinine, and lactic acid levels. Coagulopathy is evident by the elevated INR and D-dimer level and the decreased fibrinogen levels. This patient had an Hscore of 208 points (indicative of a high probability of HLH (*vide infra*)). Testing for leptospirosis, Ebstein-Barr virus (EBV) or herpes simplex virus (HSV) was not performed. The HIV screen was negative and therefore immune-reconstitution syndrome was unlikely. There was no identified new medication use making drug reaction with eosinophilia and systemic symptoms (DRESS syndrome) unlikely. A *Rickettsia typhi* serologic panel sent on D5 returned later with IgM 1:256, IgG < 1:64. *Rickettsia rickettsii* antibodies were negative. An autopsy revealed hemophagocytic histiocytes in the liver, lymph nodes, spleen, and bone marrow (Figures 1–3). With these findings, the cause of death was ascribed to HLH due to FBT.

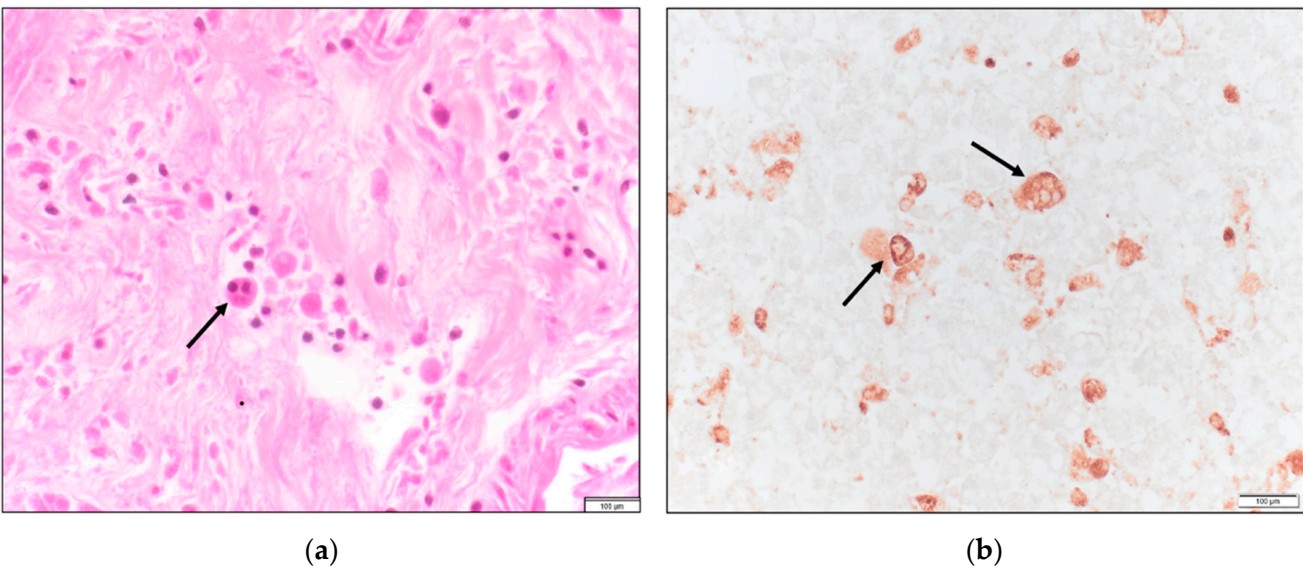

**Figure 1.** (**a**) Liver with portal hemophagocytosis demonstrated by histiocytes engulfing small lymphocytes (hematoxylin and eosin (H&E), 400×); (**b**) CD68 immunohisto-chemical stain highlights hemophagocytotic histiocytes in the liver (400×).

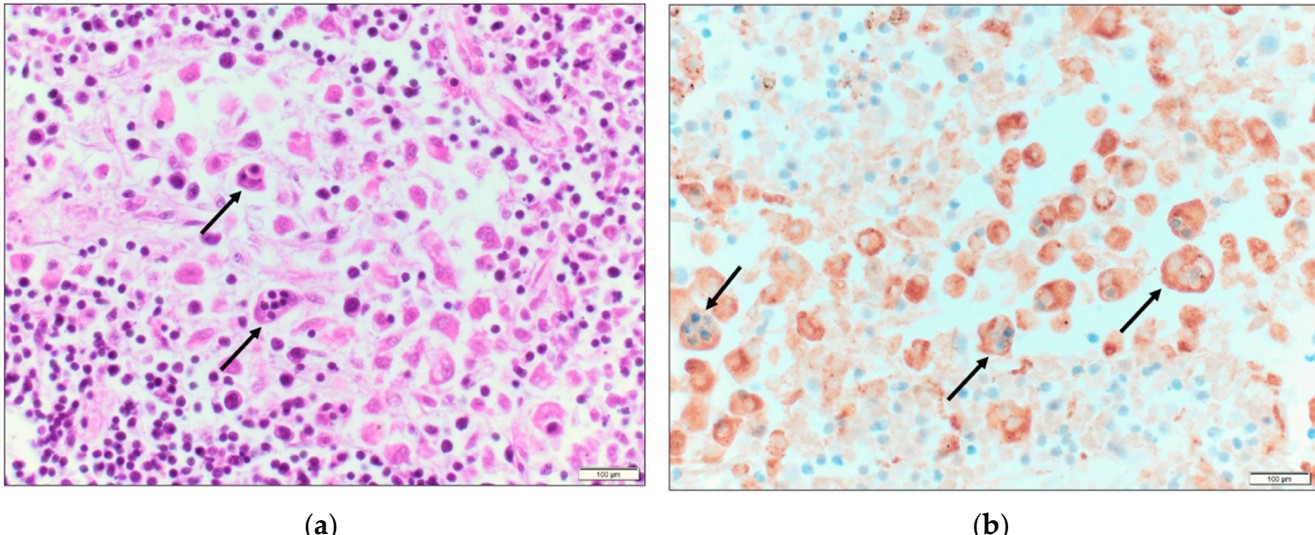

**Figure 2.** (**a**) Mediastinal lymph node with non-neoplastic activated histiocytes that exhibit hemophagocytosis (H&E, 400×); (**b**) CD68 immunohistochemical stain demonstrates hemophagocytotic histiocytes containing small lymphocytes in a mediastinal lymph node (CD68, 400×).

Other autopsy findings included acute pneumonia; bilateral pulmonary edema; a focal pulmonary infarct of the left lower lobe, emphysema, pulmonary hypertension; cardiomegaly; serous pleural and pericardial effusions; mild hepatic steatosis and centrilobular congestion; splenomegaly with congestion (300 g; average adult female spleen weight = 115 g (SD ± 51 g) [8]); and kidneys with mild global glomerulosclerosis, interstitial fibrosis, and mild arteriolonephrosclerosis. The brain showed age-related tauopathy but lacked the neuronal necrosis with focal perivascular mononuclear cell infiltrate (glial or typhus nodule) indicative of rickettsial encephalitis [9]. Although the patient was suspected to have pancreatitis based on the elevated lipase level, the pancreas was grossly normal at autopsy (a histopathologic exam was not performed). It is likely that FBT also induced the pulmonary infarction, as suggested by the high d-dimer levels.

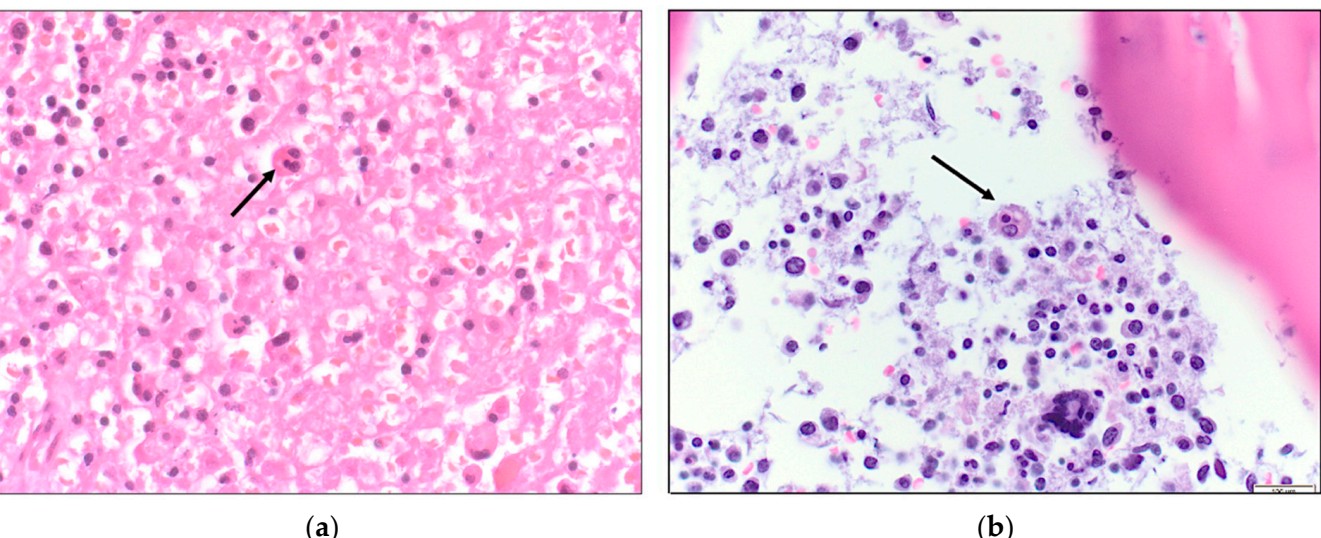

**(a)**　　　　　　　　　　　　　　　　　　　　**(b)**

**Figure 3.** (**a**) Histiocytes engulfing small lymphocytes in the spleen (H&E, 400×; (**b**) Bone marrow infiltration by non-neoplastic histiocytes that display hemophagocytosis (H&E, 400×).

## 3. Discussion

Various criteria are used to define a confirmed or probable case of FBT. Confirmed cases are defined as a clinically compatible illness with one of the following serologic findings: ≥4-fold rise in antibody titer by IFA between acute and convalescent specimens; or a single IgM or IgG titer of ≥1024 in the endemic area. Probable cases are defined as a clinically compatible illness and a single serologic titer of ≥128 by IFA [10]. The *R. typhi* titers obtained for this patient were IgM 1:256 and IgG < 1:64. Because of the rapid fatal course of infection and the slow development of detectable serologic titers in FBT [11], there was an inadequate period for the patient to develop an IgG response. The illness was clinically compatible because of the presentation with elevated transaminases and lactate dehydrogenase levels; hyponatremia; hypoalbuminemia; thrombocytopenia; and the lack of responsiveness to beta-lactam treatment. Epidemiologically, the patient resided in a county of Texas with an increasing number of cases of FBT [7]. Thus, this patient is considered a probable case of FBT.

HLH can be triggered by multiple factors including malignancy, autoimmune diseases, drug hypersensitivity, and infections (EBV, HSV, cytomegalovirus, HIV, influenza, tuberculosis, leishmaniasis, among others) [12]. The pathogenesis of HLH involves uncontrolled hyperinflammation that results in excessive (predominantly) cytotoxic CD8+ T-lymphocyte and histiocyte stimulation. Although an excess stimulation ensues, there is defective cytolysis due to inherited or acquired deficiencies in components such as perforin that assist with infected target cell elimination. The result is an endless loop of T-lymphocyte stimulation and cytokine-induced damage to organ systems [12–14]. Such a response triggered by autoimmune conditions is termed Macrophage Activation Syndrome [12]. Members of the order Rickettsiales (genera *Rickettsia*, *Anaplasma*, *Ehrlichia*, and *Orientia*) have been implicated as causative for HLH in multiple cases. A summary of 16 *Rickettsia conorii* cases (excluding those co-infected with *Coxiella*) implicated in secondary HLH was compiled by Karra et al., in addition to a single adult case due to *R. japonica* [15]. Our search finds *R. rickettsii* as a causative pathogen in three cases [16–18]. HLH has not been reported in epidemic typhus (due to *R. prowazekii*).

A 2018 systematic review of HLH in scrub typhus identified 17 adult and 13 pediatric cases. Among the 30 scrub typhus patients with HLH, two died of multi-organ failure with intracranial hemorrhage, giving a mortality rate of 6.7%. One other patient suffered permanent neurological sequelae [19]. In a 2021 study from India of 58 pediatric patients with scrub typhus, 31% of cases were complicated by HLH [20].

Of infections due to organisms of the Order Rickettsiales in adults, ehrlichiosis may cause the highest rate of HLH. The largest single series of patients with ehrlichiosis included 147 subjects infected with *E. chaffeensis* and 10 patients with *E. ewingii*. Sixteen percent of these ehrlichiosis patients suffered HLH, but the overall prognosis was favorable, with only one death [21]. In a series of 29 patients with anaplasmosis, two patients had HLH, both of whom died [22].

Thus, there is no relationship between incidence of HLH in Order Rickettsiales infections and the underlying virulence of the organism. Although HLH is well-documented in *O. tsutsugamushi* infection, which is caused by a virulent organism, it is uncommon in Rocky Mountain Spotted Fever and unreported in epidemic typhus, which are also due to highly virulent organisms. Conversely, HLH is relatively common in ehrlichiosis, due to organisms of much lesser virulence than *O. tsutsugamushi*, *R. rickettsii*, and *R. prowazekii* [16–21].

For FBT, Walter and coworkers reported three patients with HLH out of 32 cases [23]. In another series of 32 FBT patients, Miguélez et al. reported one case of HLH [24]. Tsioutis and coworkers reported only these four cases of HLH in a summary of the clinical presentation of 2074 adult and pediatric patients with FBT that were published between 1980 and 2016 [1]. Since that time, five cases of HLH ascribed to FBT have been published with case details (Table 2), in addition to the current case.

**Table 2.** Previous Cases of Hemophagocytic Lymphohistiocytosis Associated with Flea-borne Typhus.

| Case No., [Ref], Year | Age (yrs), Sex | Clinical Presentation; Laboratory Findings | Treatment | Outcome |
|---|---|---|---|---|
| 1, [25], 2022 | 61, M | Rash; Anemia, thrombocytopenia, lymphocytopenia, hyponatremia, hypoalbuminemia, hyperferritinemia, hypertriglyceridemia, and elevated creatinine, CRP, and transaminase levels; renal failure. Bone marrow aspiration found hemophagocytosis. Convalescent *R. typhi* IgG 1:1024 | Renal failure improved with hydration. sc anakinra for 5-d. Doxy was added 3 days later and given for 21 days. | Apyrexia within 12 h after starting doxy and CRP decreased within 48 h. Clinically improved. |
| 2, [26], 2019 | 39, M | Fever, nausea, vomiting, diarrhea, headache, neck stiffness for 10 days; thrombocytopenia, hypoalbuminemia, and elevated creatinine, AST, LDH, and bilirubin. Triglycerides- 397 mg/dL; ferritin- 4270 ng/mL. LP on day 3 showed glucose 39 mg/dL; protein 166 mg/dL; 36 white blood cells/µL (75% neutrophils). A bone marrow biopsy on day 4 showed hemophagocytosis. *R. typhi* assays obtained on day 2 showed IgM 1:512 and IgG 1:64. Convalescent IgG 1:1024. | One dose pip-tazo. On day 2, pt became febrile and hypoxic, and antibiotics changed to ampicillin, ceftriaxone, doxy, and vanco. Patient intermittently unresponsive and required intubation on day 3. Dexamethasone started. Doxy given for 7 wks. | Extubated on hospital day 6. Clinically improved. |

**Table 2.** *Cont.*

| Case No., [Ref], Year | Age (yrs), Sex | Clinical Presentation; Laboratory Findings | Treatment | Outcome |
|---|---|---|---|---|
| 3, [27], 2020 | 2, M | Fever, rash. Pt found to have lymphadenopathy, splenomegaly, hepatomegaly; Pancytopenia, hypoalbuminemia, hyponatremia, hypofibrinogenemia, elevated transaminase levels, and triglycerides. Bone marrow aspiration inconclusive. Coagulopathy later developed. LDH- 6700 IU/L. Another bone marrow biopsy at 2nd wk revealed hemophagocytosis. *R. typhi* serologic panel on d-14 was negative. Repeat serologic evaluation on day 27 showed IgM 1:128 and IgG 1:512. | Treated with amoxicillin and nimesulide. Pancytopenia worsened requiring multiple transfusions. During wk-3 after admission, fever persisted and doxy initiated. Methylprednisolone later started. During hospital week 4, developed petechiae and gingival bleeding. Meropenem, vanco, and amphotericin were started, Doxy was suspended after seven days due to gastric bleeding. Dexamethasone was added. | Renal and respiratory failure occurred. Mechanical ventilation and pressors were started, but diffuse alveolar hemorrhage developed. DEATH on hospital day 35. |
| 4, [28], 2014 | 52, F | Polyarthralgia, fever, rash, splenomegaly; pancytopenia, hyponatremia, hyperferritinemia, hypertriglyceridemia, elevated transaminases. Vision change, retinitis seen. Bone marrow biopsy revealed hemophagocytosis. *R. typhi* IgM 1:1024 and IgG 1:2048 | Cefazolin and gentamicin. Required transfusions of red blood cells and platelets. Doxy, corticosteroids, and IVIG were started. Doxy stopped after 7 days of apyrexia. | Apyrexia after 3 days and lab abnormalities improved after 9 days of doxy. Clinically improved. |
| 5, [29,30], 2018 | 5, F | Fever, rash, bruising, headache, cough, abdominal pain, vomiting for 6—days, tachycardia, hypotension; Thrombocytopenia, anemia, lymphopenia, hypertriglyceridemia, hyponatremia, elevated creatinine, LDH, ferritin, and transaminase levels. Developed metabolic acidosis and disseminated intravascular coagulation. Bone marrow biopsy was normal. *R. typhi* IgM 1:128 and IgG 1:1024. Convalescent IgG titer at 4 wks after presentation was 1:4096. | Started cefotaxime. Changed to vanco and pip-tazo 12 hrs after admission, with worsening hypotension and respiratory distress. Required 3 pressors, oscillatory ventilation, inhaled NO, bicarbonate, started on day 2. | Following doxy, rapidly improved (off pressors within 4 days and off dialysis and extubated within 6 days). Discharged after 14 days of vanco, pip/tazo, and doxy. |

Abbreviations: Ref, reference; CRP, C-reactive protein; sc, subcutaneous; doxy, doxycycline; IgG, immunoglobulin G; AST, aspartate aminotransferase; LDH, lactate dehydrogenase; LP, lumbar puncture; pip-tazo, piperacillin-tazobactam; vanco, vancomycin; wk, week; IgM, immunoglobulin M; pt, patient; IVIG, intravenous immunoglobulin; NO, nitric oxide; FFP, fresh frozen plasma.

The five FBT patients with HLH included three males and two females; three were adults and two were children. All five patients presented with fever, elevated transaminase levels, and hypertriglyceridemia. Other common notable findings were rash, thrombocytopenia, hyponatremia, and hyperferritinemia, which occurred in 80% of the patients. Eighty percent of cases were treated with doxycycline along with immunosuppressive medication; 80% received corticosteroids, and single cases were treated with anakinra and intravenous immune globulin. A variety of FBT complications occurred along with the HLH. Case 2 was complicated by aseptic meningitis, which has been reported in FBT [9]. Cases 3 and 5 were accompanied by coagulopathy, as previously described in FBT [2,31]. Case 4 occurred with retinitis causing symptomatic vision change, a rare presentation of FBT [32]. In regard to the autopsy findings observed in our case, pneumonia, pulmonary embolism, and glomerulosclerosis have been previously reported in FBT [33–35].

While flea-borne rickettsioses are not a well-established cause of HLH in the primary literature, a growing number of HLH cases incriminate these organisms. It is imperative

to recognize HLH early in the course of a patient's illness. An overlap with sepsis and multi-organ dysfunction is frequent and expected, making the diagnosis difficult [15]. While this is the case, halting the immune system's exuberant response to such a condition is the highest priority. Left unchecked, HLH invariably progresses to death [15]. Treatment for secondary HLH is treatment of the underlying condition, corticosteroids, and possibly intravenous immune globulin (IVIG). In imminent organ failure, etoposide initiation is recommended. Anakinra (an IL-1 receptor antagonist) and tocilizumab (a monoclonal antibody against the IL-6 receptor) have been used as well [17]. Some recommend a graded approach, with corticosteroids used first (methylprednisolone at 1 g/d for 3–5 days); then cyclosporine (2–7 mg/kg/d) or anakinra (2–6 mg/kg, up to 10 mg/kg in divided doses); and later, etoposide (50–100 mg/m2 once weekly) [12]. In the United Kingdom, the National Health Service now recommends anakinra as the drug of choice for the treatment of HLH of all etiologies, in addition to treating the underlying cause [36].

With fever, splenomegaly, bicytopenia, hypertriglyceridemia/hypofibrinogenemia, and evidence of phagocytosis, our patient met five of eight criteria, fulfilling the diagnosis, but without high suspicion for Rickettsia as the inciting trigger for HLH, this diagnosis was not initially considered, only to be uncovered with the autopsy findings discussed above, classic for the disease. While some degree of hemophagocytosis is expected in sepsis, HLH must be part of the differential diagnosis in patients with pronounced cytopenias and elevated inflammatory markers. The 2004 Histiocyte Society HLH criteria were initially implemented for use in the context of pediatric patients with primary HLH [15]. Its use is flawed for the reasons elaborated by Karra and coworkers: many of the diagnostic criteria do not have exact cut-off values and the need for biomarkers, such as interleukin-2 receptor level and natural killer cell activity for diagnosis, which are tests that are not readily available [15]. In addition, hyperbilirubinemia and transaminase levels are not part of the 2004 HLH scoring system. An alternative is the HScore which uses the following weighted variables: known immunosuppressed state; maximum temperature; hepatomegaly and/or splenomegaly; extent of cytopenias; degree of hyperferritinemia; degree of hypertriglyceridemia; hypofibrinogenemia; elevation in serum aspartate aminotransferase; and the presence of hemophagocytosis in the bone marrow aspirate. The median Hscore is 230 (interquartile range (IQR)] 203–257)) for patients with a positive diagnosis; 125 (IQR 91–150) for patients with a negative diagnosis), affording a high probability of HLH for this patient [37]. Differentiating HLH from sepsis and multi-organ dysfunction is crucial because in HLH intensive immunosuppressive therapy is necessary, as compared to the usual sepsis protocol that would be used for a critically ill patient with multi-organ dysfunction. In HLH, treatment with an immunosuppressive regimen has often proven effective and may be lifesaving. Based on the current case and our literature review, FBT should be considered in the differential diagnosis of HLH in the appropriate geographic setting.

**Author Contributions:** Conceptualization, C.L.D. and G.M.A.; writing—original draft preparation, D.C., O.F. and M.A.; writing—review and editing, C.L.D. and G.M.A.; supervision, G.M.A. All authors have read and agreed to the published version of the manuscript.

**Funding:** This research received no external funding.

**Institutional Review Board Statement:** Not applicable.

**Informed Consent Statement:** Not applicable.

**Data Availability Statement:** Not applicable.

**Conflicts of Interest:** The authors declare no conflict of interest.

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
