# Peer review of "Flea-Borne Typhus Causing Hemophagocytic Lymphohistiocytosis: An Autopsy Case"

_2036-7449, doi:10.3390/idr15010014_

Round 1

Reviewer 1 Report

The authors report the case of an elderly diabetic lady with euglycemic diabetic ketoacidosis progressing to multi-system organ failure and death, with findings of HLH on autopsy, that was thought to be triggered by underlying endemic typhus infection. They further review the literature to describe other cases related to vector-borne infections and specially related to R. typhi infection.

Overall, the case report is well written with informative pathologic images from the autopsy. The authors have also done a commendable job in reviewing the literature and summarizing information related to HLH cases that appear to have been triggered by endemic typhus infection. I have a few points that I would like to raise regarding her initial presentation, as well as other points that will help further strengthen the manuscript.

Major Comments

1. The authors have extensively described the initial laboratory abnormalities at the time of clinical presentation, but description of her initial physical examination is limited. She is reported to be lethargic, but was there concern for meningismus or altered sensorium? Was there concern for a rash or hepatosplenomegaly? If her systemic examination was unremarkable, please add a line mentioning this.

2. It would also be helpful to include certain other epidemiologic details – where the patient resided (an urban or rural area of Texas), whether she had any pets at home, if she had any known outdoors exposure or hobbies. Additionally, adding details of when the patient presented (for example, summer months of 2021/fall of 2022) will be informative, as we know that the seasonality of some of the vector-borne infections helps drive the infectious workup being performed. Lastly, I would also recommend adding details related to control of her underlying DM – it appears to have been poorly controlled (given her polypharmacy and presentation with DKA), but including a HbA1c level will be helpful.

3. Given her epidemiologic exposure to wild animals, it will be educative to know whether the patient was evaluated for other infectious triggers for HLH such as other tick-borne illnesses and leptospirosis (that can also be a potential trigger, as recently reported by Munasinghe et al DOI: 10.1155/2021/3451155). Was she also evaluated for viral infections such as EBV or HSV?

4. Pages 6 and 7 of 10, Table 2 – The study authors have summarized clinical characteristics of other reported cases of HLH that occurred due to flea-borne typhus infection. There are relevant pieces of information being conveyed in the table, but it is very difficult to follow this information. To increase readability, I would suggest modifying the data in the table into certain broad headings – Case, Reference, Year; Age/Sex; Clinical Presentation; Notable laboratory findings; Treatment/s; Rickettsial serology results; Complications; Outcome (death/alive).

Minor Comments

1. Page 2of 10, Line 65 – the study authors report that “viral hepatitis serologic screen” was negative. Was this serologic testing for hepatitis A, B and C? Please clarify this point.

2. Page 2 of 10, line 77 – the study authors report that after ICU transfer, broad-spectrum antibiotics were initiated. Please clarify which antibiotics these were.

3. Page 3 of 10, Table 1 – Did the patient have serum Ferritin measurement as well during her hospitalization? It would be helpful to add that information if available, as that is one of the other laboratory criteria used in HLH diagnosis.

4. Page 6 of 10, lines 162-166 – please include relevant references for the statements “Although HLH is well-documented in O. tsutsugamushi infection, which is caused by a virulent organism, it is uncommon in Rocky Mountain Spotted Fever and unreported in epidemic typhus, which are also due to highly virulent organisms. Conversely, HLH is relatively common in ehrlichiosis, due to organisms of much lesser virulence than O. tsutsugamushi, R. rickettsii, and R. prowazekii.”

5. Page 6 and 7 of 10, Table 2 – please include the full-form of the abbreviation of FFP in the Key (I presume it relates to fresh frozen plasma).

6. Page 8 of 10, line 224 – the study authors mention the H-score of their patient. I think this would be better suited to be initially mentioned in the Case history (lines 96-102).

Author Response

Dear Reviewer,

We would like to thank you for the time reviewing our work and the insightful recommendations that have led us to produce a better version of our original article. Below we itemize our responses and edits to your major and minor comments.

With Regards,

Christopher Dayton

Response to Major Comments of Reviewer #1:

  1. The authors have extensively described the initial laboratory abnormalities at the time of clinical presentation, but description of her initial physical examination is limited. She is reported to be lethargic, but was there concern for meningismus or altered sensorium? Was there concern for a rash or hepatosplenomegaly? If her systemic examination was unremarkable, please add a line mentioning this.

Response 1.  We have added the statement “Physical exam was otherwise unremarkable”.

  1. It would also be helpful to include certain other epidemiologic details – where the patient resided (an urban or rural area of Texas), whether she had any pets at home, if she had any known outdoors exposure or hobbies. Additionally, adding details of when the patient presented (for example, summer months of 2021/fall of 2022) will be informative, as we know that the seasonality of some of the vector-borne infections helps drive the infectious workup being performed. Lastly, I would also recommend adding details related to control of her underlying DM – it appears to have been poorly controlled (given her polypharmacy and presentation with DKA), but including a HbA1c level will be helpful.

Response 2. In the section on case presentation, “urban Texas (Bexar Co.)” was added to residential details; “spring of 2021” was added to her presentation details. “She had no pets of her own and had no outdoor activity or known hobbies” added to presentation details. The hemoglobin A1c was 7.1% and this was added to the case presentation.

  1. Given her epidemiologic exposure to wild animals, it will be educative to know whether the patient was evaluated for other infectious triggers for HLH such as other tick-borne illnesses and leptospirosis (that can also be a potential trigger, as recently reported by Munasinghe et al DOI: 10.1155/2021/3451155). Was she also evaluated for viral infections such as EBV or HSV?

Response 3. Rickettsia rickettsii antibodies were negative and this detail was added to the case.   Ehrlichiosis is a reportable condition in Texas, but it is uncommon, with only an average of 11 cases per year (vs 375/ year for flea-borne typhus (Erickson TA, et al.  The epidemiology of human ehrlichiosis in Texas, 2008-2017.  Ticks Tick Borne Dis 2012 12:101788; Texas Dept of Health Services. Flea-borne Typhus Cases in Texas by County Reported, 2008-2019. https://www.dshs.texas.gov/IDCU/disease/typhus/Typhus-2008-2019.pdf).  Furthermore, the patient’s county of residence, Bexar Co., has a very low incidence of ehrlichiosis, but is one of the most affected counties by flea-borne typhus (Erickson, ibid.; TX Dept of Health Services, ibid.)

She was not evaluated for leptospirosis, EBV or HSV. These missing detail is now mentioned in the case.  However, these potential causes of HLH seem unlikely in this patient.

There are only two reported cases of HLH associated with leptospirosis in the medical literature.  Furthermore, the patient did not present with the characteristic features of leptospirosis: conjunctivitis, icterus, myalgias, and initially elevated creatinine (DA Haake, PN Levett. Leptospirosis in humans.  Curr Top Microbiol Immunol. 2015;387:65-97). Thus, we did not consider leptospirosis in the diagnostic evaluation of HLH for this patient.

EBV infection is a common cause of HLH, but EBV-associated HLH tends to occur in young adult males. In the largest series of 133 EBV-associated HLH cases in persons older than 14-years old, the average age was 26 years old, with a 2.2:1 male:female ratio [Lai et al. Epstein-Barr virus-associated hemophagocytic lymphohistiocytosis in adults and adolescents-a life-threatening disease: analysis of 133 cases from a single center. Hematology, 2018, 23:810]

HSV can also cause HLH, but the patient had no history of prior HSV infection and no clinical evidence of a current herpes outbreak.  Furthermore, none of the organs at autopsy showed cells displaying the specific cytopathic effect of HSV (Cowdry Type A intra-nuclear inclusion bodies).    

Thus, based on these considerations, flea-borne typhus remains as the most likely instigating factor for the HLH in this patient.   

  1. Pages 6 and 7 of 10, Table 2 – The study authors have summarized clinical characteristics of other reported cases of HLH that occurred due to flea-borne typhus infection. There are relevant pieces of information being conveyed in the table, but it is very difficult to follow this information. To increase readability, I would suggest modifying the data in the table into certain broad headings – Case, Reference, Year; Age/Sex; Clinical Presentation; Notable laboratory findings; Treatment/s; Rickettsial serology results; Complications; Outcome (death/alive).

Response 4. Table 2 has been modified. The Table was difficult to read with 8 columns suggested by the reviewer.  Thus, we reduced it to: Case, Reference, Year; Age/Sex; Clinical Presentation, Laboratory findings (including Rickettsial serology results); Treatment; Outcome

Minor Comments

  1. Page 2of 10, Line 65 – the study authors report that “viral hepatitis serologic screen” was negative. Was this serologic testing for hepatitis A, B and C? Please clarify this point.

Response 1. Viral hepatitis serologic screen for hepatitis A, B and C was negative. This point has been clarified.

  1. Page 2 of 10, line 77 – the study authors report that after ICU transfer, broad-spectrum antibiotics were initiated. Please clarify which antibiotics these were.

Response 2. Vancomycin, cefepime and metronidazole were initiated. This statement has been clarified.

  1. Page 3 of 10, Table 1 – Did the patient have serum Ferritin measurement as well during her hospitalization? It would be helpful to add that information if available, as that is one of the other laboratory criteria used in HLH diagnosis.

Response 3. A ferritin level was not measured. This missing detail has now been mentioned in the report.

  1. Page 6 of 10, lines 162-166 – please include relevant references for the statements “Although HLH is well-documented in O. tsutsugamushi infection, which is caused by a virulent organism, it is uncommon in Rocky Mountain Spotted Fever and unreported in epidemic typhus, which are also due to highly virulent organisms. Conversely, HLH is relatively common in ehrlichiosis, due to organisms of much lesser virulence than O. tsutsugamushi, R. rickettsii, and R. prowazekii.”

Response 4. Relevant references supporting this statement have been included in the manuscript.

  1. Page 6 and 7 of 10, Table 2 – please include the full-form of the abbreviation of FFP in the Key (I presume it relates to fresh frozen plasma).

Response 5. Abbreviation has been expanded.

  1. Page 8 of 10, line 224 – the study authors mention the H-score of their patient. I think this would be better suited to be initially mentioned in the Case history (lines 96-102).

Response 6. The H-score was included in the case presentation section.

Reviewer 2 Report

This is a well written report regarding a patient who developed secondary HLH due to FBT and unfortunately died due to MOF. 

I think the case is interesting, novel and summary of the literature has adequate educational value. In retrospect, doxycycline should have been added sooner, as soon as, thrombocytopenia worsened and transaminitis became more severe. 

I have one major criticism and few minor comments:

Major:

1. Discussion lacks differential diagnosis section. For example- HIV testing was not reported, and the presentation of this patient is very similar to KICS ( A Fatal Case of Kaposi Sarcoma Immune Reconstitution Syndrome (KS-IRIS) Complicated by Kaposi Sarcoma Inflammatory Cytokine Syndrome (KICS) or Multicentric Castleman Disease (MCD): A Case Report and Review - PubMed (nih.gov). 

This should be discussed further. 

Minor comments: 

Line 76- "hypoxic" or hypoxemic respiratory failure?

Line 77- " broad spectrum antibiotics were initiated" should be changed to antimicrobial regimen was broaden - since patient was already on ceftriaxone which is antibacterial antibiotic of broad spectrum. Please be specific which antimicrobial was added to ceftriaxone?  

Author Response

Dear Reviewer,

We would like to thank you for the time reviewing our work and the insightful recommendations that have led us to produce a better version of our original article. Below we itemize our responses and edits to your major and minor comments.

With Regards,

Christopher Dayton

Major:

  1. Discussion lacks differential diagnosis section. For example- HIV testing was not reported, and the presentation of this patient is very similar to KICS ( A Fatal Case of Kaposi Sarcoma Immune Reconstitution Syndrome (KS-IRIS) Complicated by Kaposi Sarcoma Inflammatory Cytokine Syndrome (KICS) or Multicentric Castleman Disease (MCD): A Case Report and Review - PubMed (nih.gov).

This should be discussed further.

Response 1. HIV negative status included in the report. Differential diagnoses included in the manuscript.

Minor comments:

Line 76- "hypoxic" or hypoxemic respiratory failure?

Response. Term clarified to hypoxemic respiratory failure.

Line 77- " broad spectrum antibiotics were initiated" should be changed to antimicrobial regimen was broaden - since patient was already on ceftriaxone which is antibacterial antibiotic of broad spectrum. Please be specific which antimicrobial was added to ceftriaxone? 

Response. Antimicrobials clarified in this section.

Round 2

Reviewer 1 Report

I would like to commend the authors for making the suggested changes in their manuscript.

I do not have any further comments or suggestions.

Reviewer 2 Report

The paper has been improved.